# LLM as a Mastermind: A Survey of Strategic Reasoning with Large Language Models

**Yadong Zhang**[1][*] **Shaoguang Mao**[2][†] **Tao Ge**[2], **Xun Wang**[2], **Yan Xia**[2],
**Wenshan Wu**[2], **Ting Song**[2], **Man Lan**[1], **Furu Wei**[2]

[1]Department of Computer Science and Technology, East China Normal Uiversity
[2]Microsoft Research Asia
{yadongzhang@stu, mlan@cs}.ecnu.edu.cn, {shaoguang.mao, tage, xunwang,
yanxia, wenshan.wu,tsong, fuwei}@microsoft.com

## Abstract

This paper presents a comprehensive survey of the current status and opportunities for Large Language Models (LLMs) in **strategic reasoning**, a sophisticated form of reasoning that necessitates understanding and predicting adversary actions in multi-agent settings while adjusting strategies accordingly. Strategic reasoning is distinguished by its focus on the dynamic and uncertain nature of interactions among multi-agents, where comprehending the environment and anticipating the behavior of others is crucial. We explore the scopes, applications, methodologies, and evaluation metrics related to strategic reasoning with LLMs, highlighting the burgeoning development in this area and the interdisciplinary approaches enhancing their decision-making performance. It aims to systematize and clarify the scattered literature on this subject, providing a systematic review that underscores the importance of strategic reasoning as a critical cognitive capability and offers insights into future research directions and potential improvements.

## 1 Introduction

Large Language Models (LLMs) have ushered in a new era in artificial intelligence, particularly highlighting the potential in performing reasoning tasks, including common sense question answering(Talmor et al., 2022), and mathematical problems(Miao et al., 2021), etc.

**Strategic reasoning** (Van Der Hoek et al., 2005; Duan et al., 2024; Gandhi et al., 2023) represents a distinct art of reasoning. Generally, it involves reasonably choosing the best strategy of action in a multi-agent setting, considering how others will likely act and how one's own decisions will influence their choices. The necessity of strategic reasoning with large language models extends beyond academic curiosity; it is integral to understanding and navigating the complexities of the physical and social worlds. Human intelligence not only predicts the outcome of behaviors in the physical and social environments but also adjusts strategies based on these predictions. In order to endow AI with social attributes—making it more intellectual, responsible, and equipped with an empathetic perspective—delving into strategic reasoning with LLMs is imperative.

Strategic reasoning differentiates itself from other forms of reasoning by the **dynamism of the reasoning environment** and the **uncertainty of adversary actions**. We compare core cognitive skills required for various reasoning tasks in Table 1. It requires not only a profound understanding of the dynamic environment (context) but also the ability to make rational decisions within predictions of other participants. Strategic reasoning challenges are highly relevant to real-world issues, including business analysis and policy making.

---

[*]Work was done when interning at Microsoft Research Asia.
[†]Correspondence to: shaoguang.mao@microsoft.com

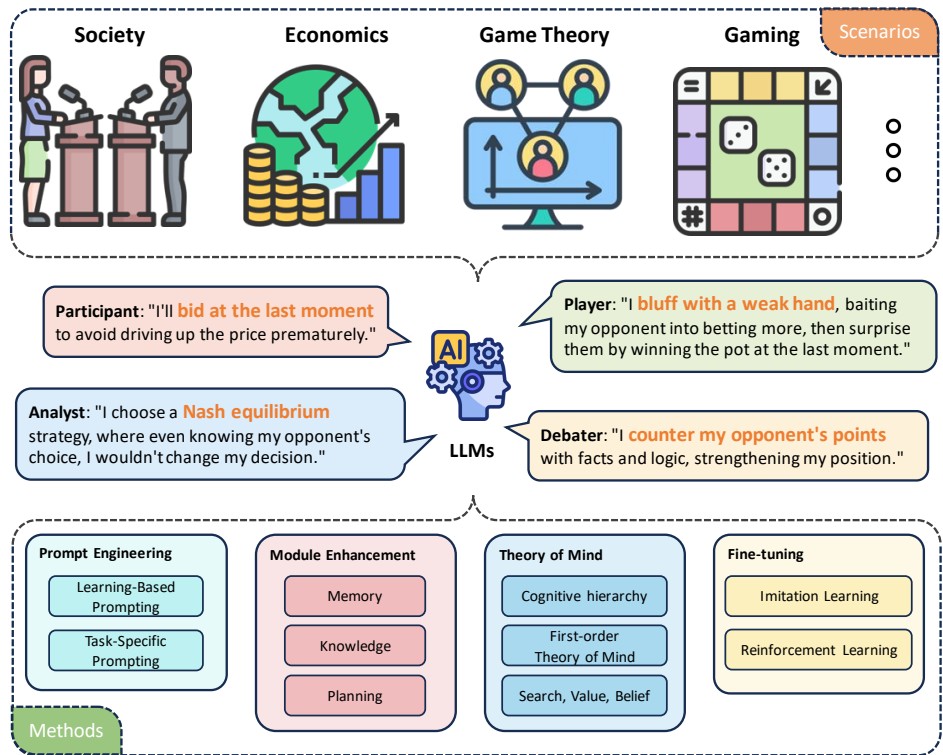

Figure 1: Strategic reasoning with Large Language Models.

Due to the intriguing characteristics, strategic reasoning has attracted increasing attention from the academic community.

Prior to the widespread adoption of large language models, strategic reasoning has been confined to intricate digitalized environments such as spatial action games, board games, and competitive video games, where agents' decision-making capabilities heavily rely on extensive simulation through reinforcement learning (Gronauer & Diepold, 2022; Arulkumaran et al., 2017; Browne et al., 2012; Silver et al., 2017). These hindered the application scope and transferability of strategic reasoning. Fortunately, the advent of LLMs has brought new opportunities for strategic reasoning. **Firstly**, the text generation capabilities of Large Language Models (LLMs) facilitate a wider range of strategic applications through the implementation of dialogue-based generative agents. **Secondly**, the powerful contextual understanding capabilities of LLMs (Ouyang et al., 2022) enable them to quickly grasp new scenarios, significantly extending the scope of AI-based strategic reasoning settings beyond the previous confines. **Lastly**, the text-based reasoning process offered by LLMs serves as a simulation of human thought (Wei et al., 2022; Kojima et al., 2022; Wang et al., 2023b), making decision-making more transparent and interpretable.

Leveraging the advantages of LLMs in decision-making and reasoning, there has been a flourishing development in enlarging scenarios recently. Meanwhile, methods from interdisciplinary fields such as cognitive hierarchy (Zhang et al., 2024c) and theory of mind (Guo et al., 2023) are being adapted to enhance the decision-making performance of LLMs. Despite the proliferation of applications and methodologies, there is a **notable absence of a systematic review** on the use of LLMs in strategic reasoning that would organize and elucidate the differences and connections among these works. Compared to the literature on multi-agent reinforcement learning (Huh & Mohapatra, 2023), utilizing LLMs for strategic reasoning significantly diverges in methodology and application scope. The review literature on using large language models for agent (Guo et al., 2024b; Wang et al., 2024; Xi et al., 2023), simulation (Gao et al., 2023) and game-playing (Xu et al., 2024b) does mention some aspects of strategic reasoning, but strategic reasoning as a critical cognitive

| Reasoning Task | Logical Deduction | Contextual Intelligence | Predictive Analytics | Abstract Thinking | Cognitive Empathy |
|---|---|---|---|---|---|
| Common Sense Reasoning | 🟡🟡 | 🔴🔴🔴 | 🟡🟡 | 🟡🟡 | 🟡🟡 |
| Mathematical Reasoning | 🔴🔴🔴 | 🟡🟡 | 🟡🟡 | 🔴🔴🔴 | 🟢 |
| Symbolic Reasoning | 🔴🔴🔴 | 🟢 | 🟢 | 🔴🔴🔴 | 🟢 |
| Causal Reasoning | 🔴🔴🔴 | 🔴🔴🔴 | 🔴🔴🔴 | 🟡🟡 | 🟢 |
| Strategic Reasoning | 🟡🟡 | 🔴🔴🔴 | 🔴🔴🔴 | 🟡🟡 | 🔴🔴🔴 |

Table 1: An analysis of common reasoning tasks and their alignment with different cognitive skills. 🟢 , 🟡🟡 , and 🔴🔴🔴 indicate low, medium, and high levels, respectively. We have not exhaustively listed all the cognitive skills related to reasoning. Instead, we have primarily selected a few representative cognitive skills associated with different reasoning tasks. The meaning of each cognitive skill is explained in Appendix A.1.

capability should be focused on and systematically analyzed. This paper aims to provide **a comprehensive overview of the state of LLMs in strategic reasoning**, shedding light on their capabilities, applications, and the road ahead for harnessing their potential more effectively.

The rest of this survey is organized in the following order: Section 2 delves into the definitions and scopes of strategic reasoning, outlining how strategic reasoning differentiates from other reasoning scenarios. Section 3 explores the applications of LLMs in strategic reasoning, categorizing tasks and application domains. Section 4 discusses existing methods to enhance LLMs in strategic reasoning, classifying approaches for employing LLMs in strategic thought processes. Section 5 discusses how to evaluate LLMs' performance in strategic reasoning, incorporating both quantitative assessments and qualitative analysis of capabilities. Lastly, Section 6 engages in a discussion on the challenges and opportunities presented by applying LLMs to strategic reasoning, offering insights into future research directions and potential improvements based on the current limitations of the research.

## 2 Definition: What is Strategic Reasoning with LLMs

Strategic reasoning can be defined as the ability to anticipate and influence the actions of others in a competitive or cooperative multi-agent setting. This involves understanding the motives, intentions, and potential actions of others, as well as the causal relationships within the environment. Unlike other forms of reasoning, which may focus on static problem-solving or single-agent decision-making, strategic reasoning is inherently dynamic and interactive, requiring a continuous assessment of the evolving situation and the intentions of other agents. In Appendix A.2, we provide a formal definition of strategic reasoning with LLMs.

The core characteristics of strategic reasoning include:

**Goal-Oriented**: The reasoning process is directed towards achieving specific objectives, often within a competitive or cooperative framework.

**Interactivity**: Strategic reasoning involves interaction among multiple agents, each influencing and being influenced by the decisions of others.

**Predictive Nature**: It requires the ability to predict the actions and responses of other agents based on limited information and uncertain outcomes.

**Adaptability**: Agents must be able to adapt their strategies in response to the actions of others and changes in the environment.

It is also important to define what falls outside our discussion's scope. Specifically, we will not address environments lacking in strategic complexity such as simulations with Generative Agents (Park et al., 2023) that do not engage in obvious strategic reasoning. Additionally, scenarios of multi-agent collaboration task solving that do not require dynamic environmental adjustments or feedback from collaborators are also excluded from the analysis of strategic reasoning. This exclusion includes both the environments and use cases

where the strategic reasoning is either absent or significantly minimized, ensuring our focus remains on the strategic applications of LLMs in contexts that demand a comprehensive understanding of goals, competition, and environmental dynamics.

## 3 Scenarios: Where to Apply Strategic Reasoning with LLMs

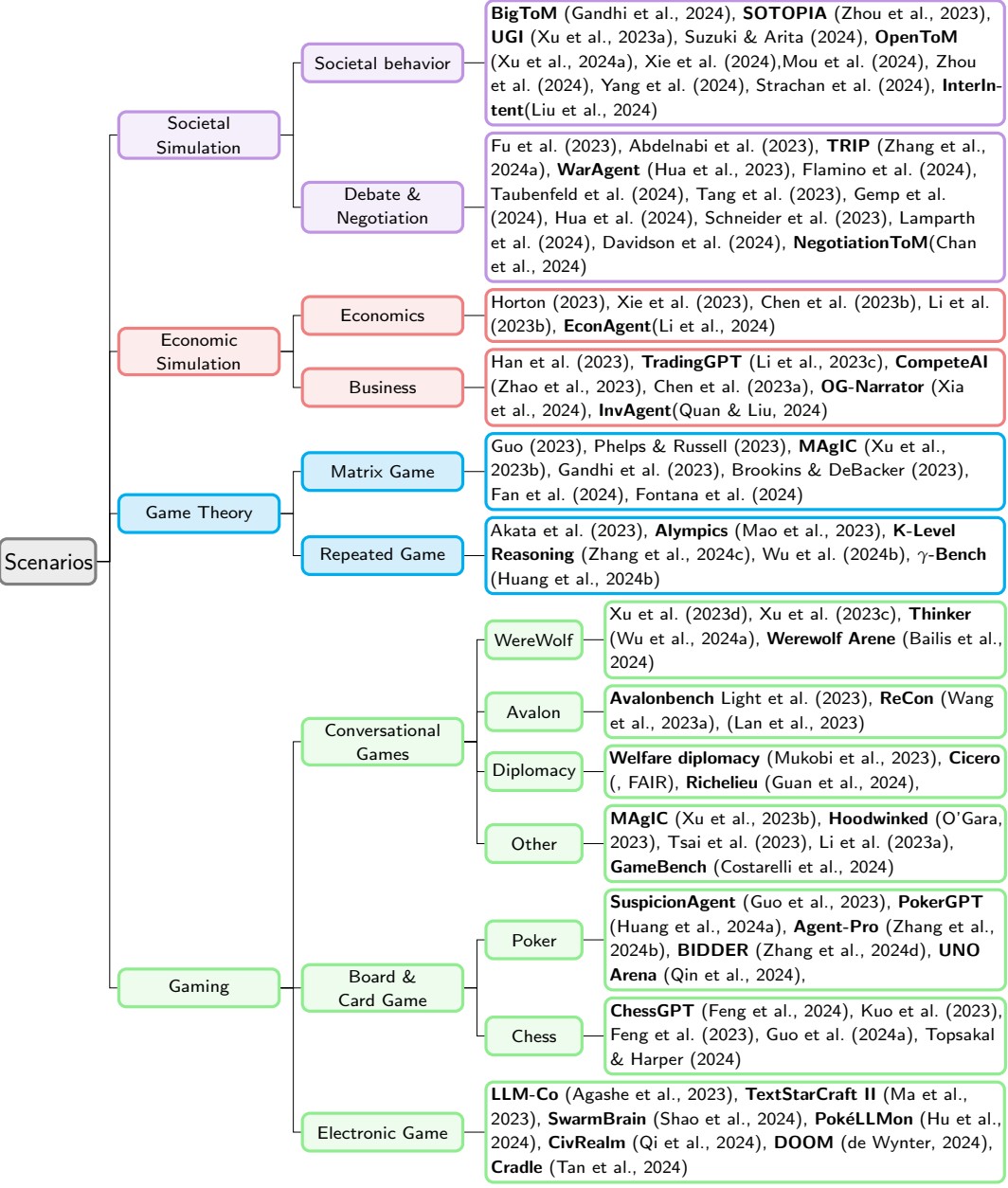

Figure 2: Taxonomy of scenarios of strategic reasoning by LLM-based agents.

This paper delineates the distinct aspects of LLM applications in strategic reasoning scenarios, illustrating how these models forecast and adapt within various settings. As shown in Figure 2, we categorize these scenarios into **societal simulation**, **economic simulation**, **game theory**, and **gaming**. Each category represents a different environment or set of conditions under which strategic reasoning is required, and together they showcase the versatility and depth of LLMs in understanding and influencing multi-agent dynamics:

**Societal Simulation** focuses on the simulation of social systems and interactions, where LLMs are used to model and predict human behavior in complex societal contexts. It involves multiple agents (individuals or groups) whose interactions are influenced by social norms, cultural values, and collective behaviors. By simulating these interactions, LLMs can help in understanding societal trends, decision-making processes, and the impact of policies or interventions. To advance the study of the social intelligence of LLMs, BigToM (Gandhi et al., 2024), SOTOPIA (Zhou et al., 2023), and OpenToM (Xu et al., 2024a) have been introduced as pivotal frameworks. These tools are engineered to assess LLMs' abilities to comprehend human psychological states as well as their social skills, such as empathy, persuasiveness (Zhou et al., 2023), and trsut (Xie et al., 2024). In the realm of political debates, Taubenfeld et al. (2024) and Tang et al. (2023) critically assess the limitations of LLMs in simulating human-like interactions, pointing out a tendency for LLM agents to adhere to inherent social biases despite attempts to engage them in diverse political perspectives. This highlights the challenges in achieving unbiased and representative simulations of societal discourse. The simulation of historical conflicts, as presented in WarAgent (Hua et al., 2023), exemplifies the potential of LLM-powered AI systems to recreate and analyze international disputes, offering a novel perspective on understanding the decisions and outcomes of major historical events such as the World Wars and the Warring States Period.

**Economic Simulation** involves modeling market dynamics, business operations, and financial decision-making processes. In this setting, LLMs are applied to understand and predict the outcomes of economic decisions, simulating scenarios like market competition, resource allocation, and investment strategies. These simulations require strategic reasoning to navigate complex economic landscapes, optimizing outcomes based on predictions of other agents' behaviors. LLMs demonstrate their capability to analyze and participate in economic systems, showcasing strategic thinking in monetary and business environments. Horton; Chen et al.; Xie et al.; Li et al. have contributed to understanding how LLM-empowered agents can simulate hiring scenarios, demonstrate rational decision-making in economic experiments, and predict stock movements. These studies underscore the capacity of LLMs to mimic realistic work and consumption decisions, potentially reshaping macroeconomic modeling. CompeteAI (Zhao et al., 2023) framework introduces a competitive environment simulated with GPT-4, focusing on the interaction between restaurant and customer agents, which illustrates the dynamics of business competition. Furthermore, AucArena (Chen et al., 2023a) demonstrates how LLMs can engage effectively in auctions, emphasizing the adaptability and strategic thinking capabilities of these models.

**Game Theory** is the study of strategic interaction among rational decision-makers. It is inherently about strategic reasoning, as it involves predicting and countering the moves of other players in various game settings. LLMs engaged in game-theoretic simulations are tested on their ability to formulate strategies in competitive, cooperative, and mixed-motive situations. This not only shows LLMs' strength in abstract strategic reasoning but also their application in practical scenarios where understanding and anticipating the actions of others are crucial. In the realm of game theory, LLMs have been instrumental in analyzing and engaging in strategic play, demonstrating their ability to model fairness and cooperation in matrix and repeated games, as highlighted in studies by Xu et al. (2023b), Gandhi et al. (2023), and Brookins & DeBacker (2023). The ongoing research into frameworks like Alympics (Mao et al., 2023) and approaches like k-level reasoning (Zhang et al., 2024c) showcases LLMs' proficiency in multi-round strategic thinking, providing insights into their long-term strategic planning capabilities.

In the context of **Gaming**, which includes board games (Feng et al., 2024; Kuo et al., 2023), card games (Guo et al., 2023; Huang et al., 2024a; Zhang et al., 2024b), and video games (Ma et al., 2023; Agashe et al., 2023; Hu et al., 2024), strategic reasoning is critical for success. LLMs are used to understand game mechanics, develop winning strategies, and adapt to opponents' tactics. This category demonstrates LLMs' ability to engage in and enhance the strategic depth of interactive entertainment, reflecting their potential to reason and make decisions in dynamic and often unpredictable environments. In conversational games like Werewolf, Chameleon, and Avalon, research by Xu et al. (2023d), Wu et al. (2024a), and Light et al. (2023) demonstrates how LLMs can enhance communication, reasoning, and deception detection among agents. In board and card games, Guo et al. (2023) and Feng et al. (2024)

have shown how LLMs can outperform traditional algorithms in poker and integrate policy learning in chess, respectively. These findings suggest a broader applicability of LLMs beyond mere simulation, potentially transforming strategic game play. Electronic gaming, including StarCraft and Pokémon, has also benefited from LLM integration. TextStarCraft II (Ma et al., 2023) and PokeLLMon (Hu et al., 2024) showcase the capability of LLMs to process game information, recommend strategies, and exhibit human-parity performance in tactical battles.

Overall, LLMs are pivotal in elucidating and enhancing strategic reasoning across diverse simulations, each category offering unique insights and challenges.

## 4 Methods: How to Improve Strategic Reasoning with LLMs

In order to enhance the performance of LLM in strategic reasoning challenges, numerous methods have recently emerged. We categorize these methods into four types based on their underlying motivations, as illustrated in Figure 3.

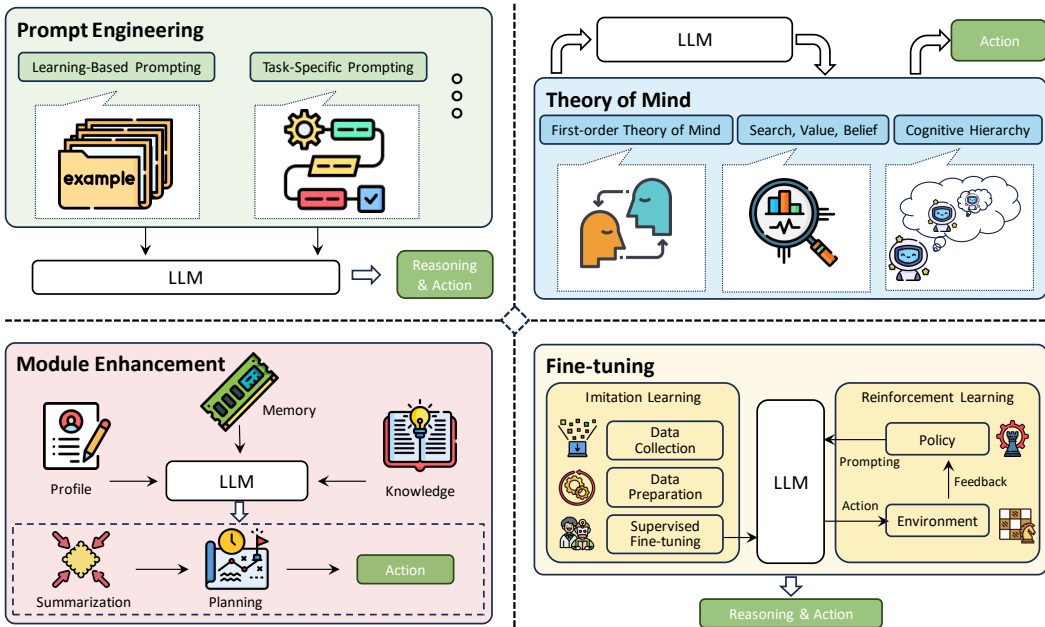

Figure 3: Methods for Improving Strategic Reasoning with Large Language Models.

**Prompt Engineering** refers to the techniques and methodologies employed in constructing effective prompts to guide Large Language Models (LLMs) in generating impactful outputs. This includes learning-Based prompting (In-context Learning (Brown et al., 2020; Wei et al., 2022)) and task-specific prompting (zero-shot Chain-of-Thought (Kojima et al., 2022)). Given the more complex contextual backgrounds of tasks involving strategic reasoning as compared to mathematical reasoning, utilizing Prompt Engineering to facilitate a clearer understanding of scenarios by LLMs presents a direct approach. To augment the situational awareness of Large Language Models (LLMs) and leverage learning from gaming history, the investigations by Fu et al. (2023), Xu et al. (2023c), Wu et al. (2023), and Hua et al. (2024) have focused on the retrieval of historical gaming data for Incontext Learning (Brown et al., 2020). These endeavors aim to enhance the proficiency of LLMs in negotiation and communication games through feedback citepfu2023improving and reflection (Xu et al., 2023c). These studies illustrate how prompt engineering not only boost LLMs' understanding and engagement in strategic games and systems but also their ability to adapt and refine these skills over time, highlighting the potential of LLMs in strategic thinking and decision-making.

**Modular Enhanced Agents** demonstrate superior performance in strategic reasoning scenarios, such as games, by integrating functionalities like memory modules for the reuse of successful strategies and leveraging external knowledge bases for the retrieval of useful information or domain-specific data. To augment the efficacy of communication and interaction in LLMs, Lan et al. (2023) proposes an innovative and encompassing framework designed for seamless adaptation to the Avalon game, including modules dedicated to summarization, analysis, planning, and action. Wthin the bargaining context, the OG-Narrator (Xia et al., 2024) incorporates a deterministic quote generator that regulates the price range of buyer propositions, along with an LLM-based narrator that formulates natural language sentences for these quotes, achieving a decuple increase in profitability relative to the baselines. In complex gaming environments, PokéLLMon (Hu et al., 2024) and Thinker(Wu et al., 2024a) address the phenomenon of illusions faced by LLM-based agents through the retrieval of external knowledge. The strategic capabilities of agents in StarCraft have been a subject of long-standing research interest. In this vein, TextStarCraft II(Ma et al., 2023) applies Large Language Models (LLMs) to StarCraft, introducing a Chain of Summarization method encompassing both single-frame summaries, aimed at processing raw observations, and multi-frame summaries, designed for the analysis of game information, provision of command recommendations, and generation of strategic decisions. This holistic enhancement of cognitive abilities makes agents more autonomous and effective in a wide range of scenarios, from simple decision-making to complex strategic reasoning and planning in dynamic scenarios.

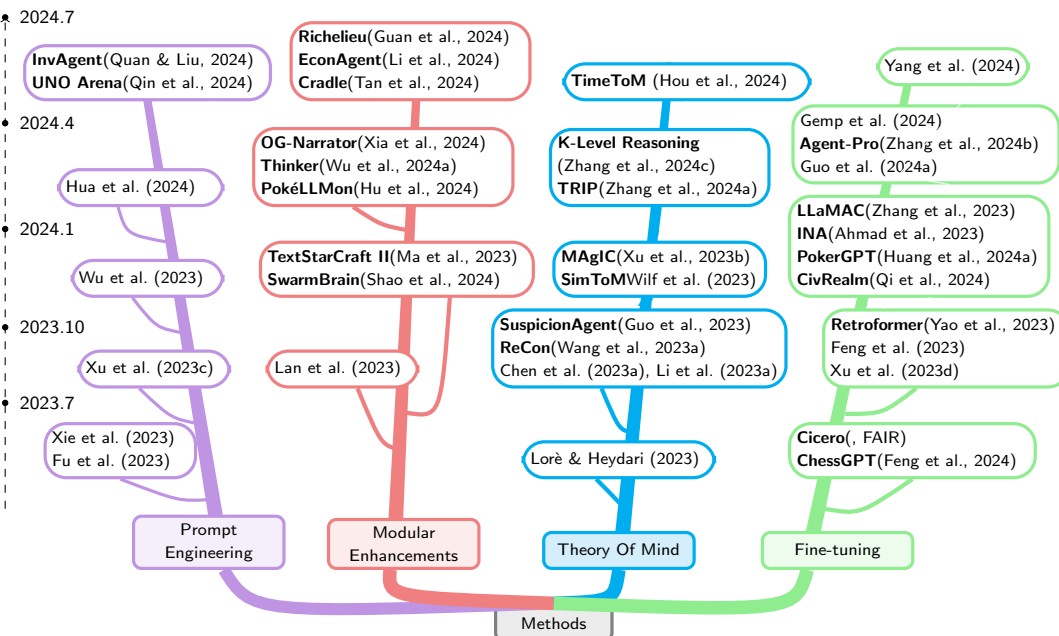

Figure 4: Overview of methods for strategic reasoning work based on LLMs.

**Theory of Mind (ToM)** is a crucial concept in strategic reasoning, enabling agents to anticipate and strategize based on the mental states of others. Gandhi et al. (2023) and Suspicion-Agent (Guo et al., 2023) employ the ToM framework for decomposing the strategic reasoning process into search algorithms, value assessments, and belief tracking environments, tailored specifically for Matrix Games and Poker, respectively. This method significantly elevates the decision-making prowess of Large Language Models (LLMs). SimTom(Wilf et al., 2023) and K-Level Reasoning (Zhang et al., 2024c) demonstrate that predictions about opponents' behaviors become markedly more precise when utilizing opponent-specific sessions. K-Level Reasoning further elucidates that a broader historical record of opponents' actions can enhance prediction accuracy, illustrating the dynamic adaptability of LLMs. This adaptability has notably improved LLMs' intelligence in DOOM (de Wynter, 2024). Additionally, Li et al. (2023a) finds that LLMs exhibit ToM capabilities in cooperative tasks, and their performance levels are comparable to reinforcement learning baselines in these

tasks. Together, these works illustrate the significant role ToM plays in enriching LLMs' strategic reasoning capabilities, demonstrating its potential to revolutionize decision-making processes across various domains.

The fusion of **Imitation Learning** and **Reinforcement Learning (RL)** with LLMs also marks a significant advancement in strategic reasoning capabilities. The initiatives of Feng et al. (2023), Guo et al. (2024a), and ChessGPT (Feng et al., 2024) are pivotal in integrating Large Language Models (LLMs) within the chess domain. To augment LLMs' chess performance, a bifurcated approach is adopted: firstly, by emulating the experiential wisdom of human players, thereby imbibing the sophisticated strategies and tactical decisions inherent to expert gameplay; and secondly, by harnessing LLMs' inherent pre-trained reasoning prowess as a value function to directly boost their operational efficacy. Gemp et al. (2024) offers a broader perspective by conceptualizing dialogue processes as game-theoretical constructs. Herein, the reinforcement learning framework is deployed to refine LLM performance in intricate interactive contexts such as meeting scheduling and public debates. Collectively, these advancements underscore the utility of incorporating imitation and reinforcement learning into LLMs, showcasing their potential to navigate and reason within complex decision-making landscapes with unprecedented sophistication.

It is important to note that the boundaries between the above methodological categories are not entirely orthogonal. For instance, methods pertaining to Theory of Mind can be implemented through prompt engineering, yet their essence lies in leveraging game theory principles to enhance LLM performance, rather than merely providing examples to refine LLM's understanding of task definitions. Finally, Figure 4 offers an overview of the methods employed to improve the efficacy of LLMs in strategic reasoning tasks.

## 5 Evaluations: How to Assess Strategic Reasoning with LLMs

The Evaluation in strategic reasoning includes **measuring outcomes** in controlled environments, where the efficacy of a model can be gauged through its performance metrics like win rates (Qiao et al., 2023), survival rates (Mao et al., 2023), and rewards. Researches such as GTBench (Duan et al., 2024) and LLMArena (Chen et al., 2024), with their sophisticated scoring systems like the Normalized Relative Advantage (NRA) and TrueSkill (Herbrich et al., 2006), respectively, provide a structured framework for this analysis. These tools not only quantify success but also allow for comparisons across various game types and difficulty levels, offering a comprehensive view of an LLM's strategic prowess.

The evaluation in strategic reasoning with LLMs also includes a **Quantitative Analysis** of the reasoning processes. Metrics that target the processes within games focus on assessing the LLM's ability to **perceive**, **predict**, and **adapt** to the dynamic environment and opponents' strategies. For instance, MAgiC (Xu et al., 2023b) evaluates the accuracy of LLMs in analyzing opponents' moves under conditions of incomplete information, and K-Level Reasoning (Zhang et al., 2024c) assesses the precision in predicting behaviors based on public information. Process-oriented evaluations are critical in multi-agent environments where nonstationarity, due to uncertain opponent behavior, significantly impacts performance. Accurate predictions of opponent behavior are essential to mitigate the effects of this nonstationarity, offering a more clear view of an LLM's strategic capabilities.

Moreover, considering the intrinsic advantages of LLMs, such as their ability to generate reasoning processes, provides a unique angle for evaluating strategic reasoning. Unlike reinforcement learning methods that focus merely on outcomes, LLMs offer explainability by detailing the inferential steps they take. This characteristic enables a more focused assessment, where the model's outputs themselves can be analyzed to understand the decision-making process better. Therefore, it is imperative to integrate these insights into the quantitative evaluation of LLMs.

**Qualitative Evaluation** shifts towards understanding the underlying mechanics of strategic reasoning in LLMs, encompassing capabilities like **deception**, **cooperation**, **discernment**, and so on. These aspects are crucial for navigating the intricacies of multi-agent interactions, where the effectiveness of a strategy is often contingent on the dynamic and often unpre-

dictable nature of opponent behavior and game states. For example, in games like Werewolf (Xu et al., 2023c) or Poker (Guo et al., 2023), the ability to bluff or cooperate effectively is as indicative of strategic reasoning as the ultimate game outcome.

The interplay between quantitative and qualitative evaluations is crucial for a holistic understanding of LLMs' strategic reasoning capabilities. While quantitative analysis offers objective benchmarks, qualitative insights expose the strategic depth and adaptability of LLMs in complex, real-world scenarios. This dual approach not only enhances the robustness of the evaluation framework but also metigates the challenges inherent in measuring cognitive processes in strategic reasoning.

The selection of these quantitative and qualitative metrics is intricately linked to specific contextual scenarios. For instance, in survival games, metrics such as agent survival rates are pertinent, whereas, in games without elimination, metrics like win rates or rewards are more relevant. Additionally, different scenarios necessitate distinct focuses on the reasoning capabilities of the employed strategies. Regarding the rationale behind selecting these quantitative metrics, we aimed at two primary objectives:

- **Rationality Assessment of LLMs**: These metrics evaluate the rationality of LLMs, highlighting the disparity between LLMs and other methods. Moreover, these metrics provide a benchmark for comparing the relative rationality of different methods.
- **Enhanced Transparency in Decision-Making**: As strategic reasoning with LLMs aims to make decision-making more transparent and interpretable, these metrics should analyze the performance of LLMs in the process of strategic reasoning.

## 6 Discussions: An Outlook on Strategic Reasoning with LLMs

### 6.1 Can the LLM agents really simulate human strategic reasoning?

Although LLMs and LLM agents have been applied in a variety of strategic reasoning scenarios, with some studies claiming the emergence of human-like intelligence capabilities in certain simulations, we argue that **there is a lack of systematic and rigorous research into what level of LLM can be employed to simulate tasks of varying complexity and cognitive difficulty in strategic reasoning**. This absence of systematic and rigorous study has led to a gap in understanding the scalability and limitations of LLMs in these contexts. Specifically, it remains unclear how different sizes and configurations of LLMs correlate with their ability to handle the decision-making and predictive tasks required in complex strategic environments. Without this knowledge, the application of LLMs in strategic reasoning risks being haphazard, potentially overlooking critical insights into the model's capabilities, decision-making processes, and potential biases or shortcomings. Therefore, a more structured approach to researching and categorizing LLM competencies in strategic reasoning is essential for fully leveraging their potential and ensuring responsible development and deployment in multi-agent strategic simulations.

### 6.2 Bridging the Divide: The Urgent Need for Unified Benchmarks

A key challenge in strategic reasoning is the absence of unified benchmarks. While recently there are some benchmarks (Xu et al., 2023b; Duan et al., 2024; Chen et al., 2024) derived from classical game theoretical problems for strategic reasoning, the vast application range of strategic reasoning, from business strategy to complex system simulations, has led to a proliferation of customized solutions focused on novel scenarios rather than deep exploration within well-defined benchmarks. This trend hinders direct method comparisons and stifles progress under a common standard. Also, as mentioned in Section 5, in tasks of strategic reasoning, it is often necessary to use a combined quantitative and qualitative evaluation approach to comprehensively evaluate the performance of LLMs both in the reasoning process and the outcome, which presents challenges for the design of a unified benchmark. **The strategic reasoning community urgently needs to collaborate on creating a suitable difficulty level, recognized benchmarks that cover its diverse applications.** Specifically, we believe that a unified benchmark should possess the following characteristics:

- **Well-Controlled Environments**: The benchmark should operate within controlled environments with minimal randomness, facilitating reproducibility and ease of experimentation.

- **Reasonable (Objective or Subjective) Evaluation**: As discussed in section 5, a unified benchmark for strategic reasoning with Large Language Models (LLMs) should be capable of objectively or subjectively assessing their strategic reasoning abilities. This includes evaluating long-term planning capabilities and dynamic adaptability, among other facets of strategic reasoning.

- **Scalability Across Scenarios**: The benchmark should be adaptable to a broad spectrum of strategic reasoning scenarios, enabling researchers to explore diverse applications and challenges.

These benchmarks would facilitate algorithm performance assessment, method comparison, and drive innovation by defining clear metrics, representative datasets, and evaluation protocols. Such efforts could unify the field, enhance knowledge sharing, and accelerate technological development.

### 6.3 Strategic Reasoning: Challenging yet Promising for Large Language Models

Strategic reasoning presents a unique challenge in LLMs. These models, which rely on next token prediction during pre-training stage, are adept at learning patterns from vast amounts of static textual data (Sap et al., 2022) but struggle to inherently grasp the subtlety of strategic reasoning. This limitation stems from the fact that strategic reasoning requires understanding complex, dynamic interactions between multiple agents, something that is not directly inferable from static text data alone. Despite this, the massive volume of data used to train LLMs allows them to model a wide range of behaviors and scenarios, indirectly capturing elements of strategic thought. By crafting prompts or algorithms that frame a problem within a strategic context, these models can generate responses that reflect strategic considerations.

However, the question remains whether scaling up—merely increasing the number of parameters and the volume of training data for a general-purpose LLM—alone could suffice for general-purpose LLMs to fully master strategic reasoning. While larger models can capture more fine and complex patterns, strategic reasoning fundamentally involves understanding intentions, predicting future actions based on those intentions, and dynamically adjusting strategies in response to evolving situations. These aspects are not purely a function of model size or data quantity. We speculate that even the most powerful general-purpose LLMs in may fall short of fully realizing strategic reasoning capabilities.

## 7 Conclusion

In conclusion, our review highlights the pivotal role of LLMs in strategic reasoning, showcasing their evolution and significant advantages in complex decision-making across various domains. Future efforts should focus on interdisciplinary collaborations to bridge theoretical advancements and practical applications, enhancing decision-making processes and strategy development. As we advance, the exploration and refinement of LLMs promise to offer substantial advancements in artificial intelligence, opening new pathways for solving complex problems and enriching strategic decision-making in an interconnected world. This calls for a concerted effort from researchers and practitioners to unlock the transformative impact of LLMs on strategic reasoning.

## Acknowledgments

We would like to express our gratitude to our colleagues at Microsoft (Jindong Wang, Adrian de Wynter) and East China Normal University (Li Cai and Xinshu Shen) for their valuable internal discussions and feedback. We also appreciate the reviewers' thorough review and valuable comments.

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

# A  Appendix

## A.1  Cognitive Skills of Reasoning

**Logical deduction** refers to the capacity to derive conclusions from premises through the application of explicit logical rules(Johnson-Laird, 1999). This mode of reasoning typically adheres to the principles of formal logic, including deductive and inductive reasoning. Deductive reasoning involves a process from the general to the specific, where conclusions are drawn based on universal truths. Inductive reasoning, conversely, is the process from the specific to the general, where general conclusions are inferred from specific observations.

Logical deduction necessitates the ability to identify and apply logical relationships, such as causality, equivalence, and contradiction.

**Contextual intelligence** denotes the ability to comprehend and interpret information within a specific context or background. It involves recognizing and interpreting the context, social norms, and implied meanings. This ability requires capturing subtle cues within a given context and understanding the significance of conversations, events, or texts. Contextual Intelligence is indispensable for language comprehension, empathetic resonance, and social interaction.

**Predictive analytics** refers to the capacity to forecast future events or trends based on existing information(Siegel, 2013). This includes analyzing data, identifying patterns and trends, and utilizing this information to make informed predictions. Predictive ability demands the integration of past and present information, employing probabilistic and statistical methods, and reasoning about possible future scenarios.

**Abstract thinking** is the ability to understand concepts, principles, and models beyond concrete and direct experiences(van de Vijver & Willemsen, 1993). This type of thinking involves generalization, categorization, and conceptualization capabilities, enabling individuals to identify similarities and differences across different situations and apply broad principles to solve problems. Abstract thinking is crucial for innovation, theoretical development, and complex problem-solving.

**Cognitive empathy** is the skill of understanding how others think and feel(Shamay-Tsoory et al., 2009). Cognitive empathy includes several aspects: 1. Perspective-taking: Naturally seeing things from someone else's point of view; 2. Fantasy: The ability to connect with fictional characters as if they were real; 3. Tactical empathy: Intentionally using perspective-taking to achieve specific goals; 4. Emotion regulation: The capability to empathize with others' emotions without becoming overwhelmed by them.

## A.2 Symbolic System of Strategic Reasoning

### A.2.1 Formulation of Strategic Reasoning Environment

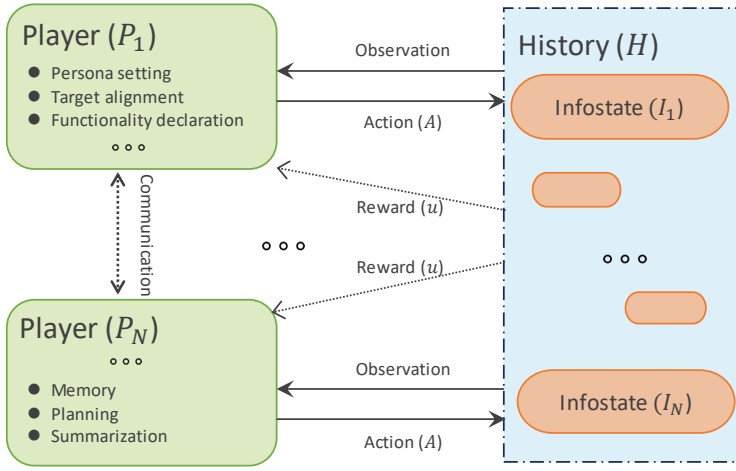

Figure 4: Environment in strategic reasoning of multi-agent systems (MAS).

We term a strategic reasoning environment as a 'GAME'. Formally, a GAME is a tuple $\langle \mathcal{N}, \mathcal{A}, \mathcal{H}, \mathcal{Z}, u, \mathcal{I} \rangle$, where:

- $\mathcal{N} = \{1, 2, ..., n\}$ is a set of $n$ agents. In the initialization of a LLM based agent participant, system message is required for configuration, commonly including persona

setting, target alignment, and functionality declaration. The system messages are passed to the LLM in the form of messages to influence the LLM's performance.

- $\mathcal{A}$ is a action set agents may take. It is a global set of state-independent actions; generally, only a subset of legal actions are available to each player at each decision point. The actions for LLMs essentially refer to the LLMs textual response given dialogue histories or prompts. In conversational environments, such as debate scenarios, any output from the LLM is considered as actions. Whereas, in scenarios with a defined finite action set, like Voting, Bidding, Poker, etc., the LLM's output needs to be parsed into a legitimate functional space.

- $\mathcal{H}$ is a set of game history. A historical record is a sequence of actions (including chance node "actions" or outcomes) taken from the start of the game. In environments based on LLM, the historical information consists of the union of dialogue histories of all players.

- $\mathcal{Z} \subseteq \mathcal{H}$ is a set of terminal histories that each represent a finished (completely played) game.

- $u : \mathcal{Z} \rightarrow \Delta_u^n \subseteq \mathfrak{R}^n$, where $\Delta_u = [u_{min}, u_{max}]$ is a utility (or payoff) function assigning each player a payoff at the end of the game, and $u_{min}$, $u_{max}$ are lower and upper bounds on those payoffs.

- $\mathcal{I}$ is a set of information states. In general, $\mathcal{I}$ is a partition of $\mathcal{H}$ such that each $i \in \mathcal{I}$ contains histories. Decisions are made by players at these states. In LLM-based environments, each player's infostate is the observable dialogue history along with their own action history and private information.

## A.2.2 Target

In the realm of strategic reasoning, we explore an environment populated by multiple agents, each endowed with the capability to engage in sophisticated reasoning processes. These agents navigate the environment with the objective of fulfilling their individual goals. Through the application of strategic reasoning, each agent assesses the potential outcomes of various actions, considering not only their own objectives but also the possible actions of other agents within the same environment. This intricate dance of decision-making and anticipation allows each agent to select the actions most likely to lead to the achievement of their goals.

Let $u_i(s)$ denote the expected utility for player $i$ under strategy profile $s$. In strategic reasoning, a player aims to maximize their expected utility, which can be symbolically represented as:

$$\max_{s_i \in S_i} E[u_i(s_i, s_{-i})]$$

where $s_{-i}$ represents the strategies of all other players except player $i$, and $E[\cdot]$ denotes the expectation operator, based on player $i$'s beliefs about the others' actions.

## A.3 Strategic Reasoning with Large Language Models vs. Reinforcement Learning

In strategic reasoning, LLMs and Reinforcement Learning (RL) represent distinct yet complementary approaches. LLMs excel in generating coherent language and leveraging extensive knowledge, making them ideal for complex problem-solving that requires creativity and deep understanding, such as in business strategy formulation or geopolitical analysis. RL, in contrast, thrives on learning optimal actions through trial-and-error interactions with the environment, suited for scenarios demanding dynamic decision-making, like autonomous systems and game optimization. However, both methodologies face challenges: LLMs may inherit biases from their training data, while RL's effectiveness hinges on precise reward definitions and environmental modeling. The future might see synergistic models that blend LLMs' comprehensive knowledge handling with RL's adaptive decision-making, promising enhanced strategic reasoning across diverse and complex scenarios.

| Criteria | Large Language Models | Reinforcement Learning |
|---|---|---|
| Knowledge Base | Extensive from diverse datasets | Acquired from specific environments |
| Contextual Understanding | Excellent in language-based tasks | Limited to specific state spaces |
| Decision-making | Abstract, human-like reasoning | Numerical, based on rewards |
| Transparency | High with text explanations | Low, often a black box |
| Flexibility | Adaptable to various scenarios | Tailored to specific tasks |
| Generalization | Transfers knowledge well across domains | Limited to trained environments |
| Interactivity | Suitable for dialogue and negotiation | Optimizes actions in environments |
| Implementation Complexity | Lower, uses pre-trained models | Higher, needs reward system design |
| Real-time Adaptation | Limited without updates | Excellent in dynamic environments |

Table 2: A Comparison of Strategic Reasoning with Large Language Models vs. Reinforcement Learning

