# OpenReview forum: "LLM as a Mastermind: A Survey of Strategic Reasoning with Large Language Models"
_colmweb.org/COLM/2024/Conference — COLM_

### Official Review · Reviewer_dKAH · 2024-05-08

**Rating:** 6
**Confidence:** 4
**Ethics Flag:** 1

**Summary:**

This paper provides a comprehensive survey about the field of strategic reasoning with LLMs.
The authors give a clear definition of strategic reasoning, and core characteristics compared to other reasoning types.
They categorize and systematically review the scenarios applying strategic reasoning with LLMs, existing methods enhancing the LLMs' capability for strategic reasoning, and the evaluation in strategic reasoning with LLMs.
The challenges of strategic reasoning with LLMs are discussed in the end.

**Questions To Authors:**

Typos:
- Mind the space before citations.
- public debatesCollectively

**Reasons To Accept:**

1. This is a timely survey for strategic reasoning with LLMs, which is clearly an important research topic.

2. The survey is well-organized and clearly written. I especially like the formalism of the definition of strategic reasoning.

3. A broad range of works related to strategic reasoning with LLMs are well placed.

**Reasons To Reject:**

1. The categorization of scenarios only distinguishes topic differences and does not discuss how they uniquely test the reasoning capabilities of LLMs or how they present distinct features and challenges compared to other scenarios.

2. In the first few sections, the survey could benefit from a deeper analysis of the related works rather than just listing them. It would be valuable to see a more detailed examination that connects different studies, evaluates their contributions critically, and outlines a developmental roadmap of the research area. This is essential for readers to grasp the progression and interplay of the works within the field.

3. While the discussion and invitation for future contributions from the community is appreciated and well-timed, it would be beneficial to include the authors' perspectives on practical approaches to realizing them.

---

> ### Author Rebuttal · Authors · 2024-05-28
>
> Thank you for your valuable feedback and encouraging comments. We would like to take this opportunity to clarify the raised questions.
>
> **Response To Question "How scenarios uniquely test the reasoning capabilities of LLMs?"**
>
> We acknowledge the necessity for a detailed discussion regarding specific tasks within each scenario, and correspondingly discussed the advantages of LLMs in various scenarios, such as Societal Simulation, where "LLMs are utilized for the modeling and prediction of human behavior within intricate societal frameworks", and "LLMs are applied to understand and predict the outcomes of economic decisions, simulating scenarios like market competition, resource allocation, and investment strategies" for Economic Simulation.
>
> The further dicussion will cover the high-level capabilities of LLM required for different scenarios, such as planning, problem decomposition, and world modeling [1], and it will facilitate more accurate assessments of LLM strengths and limitations.
>
> **Response To Criticism "The survey could benefit from a deeper analysis of the related works rather than just listing them."**
>
> We appreciate your suggestion on a deeper analysis of the related works. We have made efforts in this regard. For instance, in Section 4 (Methods), we categorized the methods based on their underlying motivations and implementations. Additionally, we utilized a timeline (Figure 4) to delineate the developmental roadmap of these methods, providing readers with a clearer developmental trajectory.
>
> **Response To Concern "It would be beneficial to include the authors' perspectives on practical approaches to realizing them"**
>
> As mentioned in 6.2, the establishment of a universal and standardized benchmark is highly needed. Such a benchmark should adhere to three principles: **Well-Controlled Environments**, **Reasonable Evaluation**, and **Scalability Across Scenarios**. From a practical standpoint, we consider Welfare Diplomacy [2] a good initiative. This game, which incorporates cooperative and deceptive elements typical of multi-agent systems, stems from classic diplomacy games [3]. However, there is a need for the development of more quantitative metrics to evaluate the strategic reasoning abilities in such scenarios.
>
> [1] Reasoning with Language Model is Planning with World Model.
>
> [2] Welfare diplomacy: Benchmarking language model cooperation.
>
> [3] Human-level play in the game of diplomacy by combining language models with strategic reasoning

---

> > ### Comment · Reviewer_dKAH · 2024-06-04
> >
> > Thank you for your response! I will retain my original score.

---

### Official Review · Reviewer_3S3R · 2024-05-09

**Rating:** 7
**Confidence:** 4
**Ethics Flag:** 1

**Summary:**

This survey analyzes the problem definition, solutions, and evaluation of LLM strategic reasoning from various perspectives. It also provides future directions for research in the field of LLM strategic reasoning. I really like this paper and recommend it to be accepted.

**Questions To Authors:**

I do not have questions but I recommend adding more related work:
[1] TimeArena: Shaping Efficient Multitasking Language Agents in a Time-Aware Simulation
[2] AutoGen: Enabling Next-Gen LLM Applications via Multi-Agent Conversation
[3] SCIENCEWORLD: Is your Agent Smarter than a 5th Grader?

**Reasons To Accept:**

1. This paper is clearly written and easy to follow.
2. The perspective proposed in this paper is excellent. Strategic reasoning is a crucial issue within LLMs, serving as an important dimension for evaluating LLM capabilities and a significant indicator for achieving AGI
3. The classification basis for the method of strategic reasoning proposed in this article is very reasonable, and the literature review is conducted with great thoroughness.
4. In the discussion of future work, the authors note that we still lack studies on the boundaries of strategic reasoning capabilities in large models, which is indeed a crucial point.

**Reasons To Reject:**

1. A minor critique is the lack of discussion on the conditions under which quantitative and qualitative evaluation metrics are applicable when analyzing the evaluation, specifically why these particular metrics were chosen.

---

> ### Author Rebuttal · Authors · 2024-05-28
>
> Thank you for affirming our work and providing valuable suggestions regarding the existing issues.
>
> **Response to the Concern "The lack of discussion on the applicable conditions of metrics and explanation for the selection of metrics".**
>
> The selection of these metrics is intricately linked to specific contextual scenarios. For instance, in survival games, metrics such as agent survival rates are pertinent, whereas, in games without elimination, metrics like win rates or rewards are more relevant. Additionally, different scenarios necessitate distinct focuses on the reasoning capabilities of the employed strategies.
>
> Regarding the rationale behind selecting these quantitative metrics, we aimed at two primary objectives:
>
> 1. **Rationality Assessment of LLMs**: These metrics evaluate the rationality of LLMs, highlighting the disparity between LLMs and traditional methods like reinforcement learning. This comparison helps showcase the differences in rationality between LLMs and conventional approaches. Moreover, these metrics provide a benchmark for comparing the relative rationality of different methods.
> 2. **Enhanced Transparency in Decision-Making:** As discussed in our introduction, LLMs aim to make decision-making more transparent and interpretable. Therefore, these metrics analyze the performance of LLMs in the process of strategic reasoning.
>
> We greatly appreciate your additional references, which undoubtedly enhance the quality of our work.
>
> Once again, thank you for your valuable feedback and constructive recommendations.

---

### Official Review · Reviewer_5o6Z · 2024-05-10

**Rating:** 6
**Confidence:** 3
**Ethics Flag:** 1

**Summary:**

In this paper, the authors give a comprehensive review of recent work on using LLMs in strategic reasoning environments (e.g., gaming environments, economic simulation). This area of research has gained much traction in the last year (e.g., all of the papers they cite in Figures 2 and 4 involves work done exclusively in 2023 and 2024) and is integral to new applications of LLMs in neighboring disciplines such as economics, business and many others. I therefore think that a survey paper of this kind is very timely and important and I generally found their paper to be easy to read and accessible.

As with any broad survey article, certain liberties were taken with how to classify different work and concepts. For example, they exclude generative agents [Park et al] and work like this as being out of scope given that the lack of focus on goal-oriented task modeling (I’d like to see more discussion of this). They also attempted to quantify the different types of reasoning skills involved with strategic reasoning tasks versus other tasks and I found this part to be overly subjective without further discussion. Given that virtually all of this work involves prompt-based LLMs, distinguishing between “prompt-engineering” approaches versus other approaches, virtually all of which involve prompting, to be a bit confusing (in fairness, the authors point this out when they write that “it is important to note that the boundaries between the above methodological categories are not entirely orthogonal”.)

Given that this article is only a survey article and offers no new empirical or technical results, I don’t have much to criticize outside of some presentation-specific issues that I note below. Baring these concerns, my main concern is about whether a survey paper fits within the COLM technical track (I don’t think it does, I will say more below. I am willing to be convinced otherwise if other reviewers and the area chair(s) are not concerned about this).

**Reasons To Accept:**

- A comprehensive and very easy to read review of LLMs and strategic reasoning, which is motivated by the considerable interest in the topic in the last year. I think it provides a useful overview of this emerging area and that researchers will cite it for this reason.

**Reasons To Reject:**

- No new technical or empirical results; not original research but a survey article. Given the call for papers (https://colmweb.org/cfp.html) and the reviewer guidelines (https://colmweb.org/ReviewGuide.html, in particular, the stated goal of having a “technical deep, exciting, forward-looking, insightful and impact program”), I don’t the paper meets the criterion for inclusion in COLM.

- (a complicated criticism) While their review is comprehensive, it fails, in my mind, to communicate what the big problems are in the different application areas where strategic reasoning has been investigated. For example, in the scenarios involving economics or game theory, what are the big or open problems in these fields that motivate using language models in place of traditional tools? What are the prospects of success in using LLMs for these problems? Of these problems, are there particular sets of problems that the LLM field has tended to focused on, or ones that people are not paying attention to?

A lot of the discussion along these lines is left vague (e.g., they end the “scenarios” section by saying: “each category [or application area of LLMs] offer[s]… unique insights and challenges”. I was really keen to get deeper into these challenges). I think it would benefit the LLM community to know a bit more about what the core problems are in these different to better motivate interest in strategic reasoning. It might also make the article easier for others outside of LLMs to engage with this work.

---

> ### Author Rebuttal · Authors · 2024-05-28
>
> Regarding your main concern, "**whether a survey paper fits within the COLM technical track,**" we have inquired with the Program Chairs:
>
> > Q: As this is the first COLM, we seek your clear guidance on whether a survey paper is appropriate for the COLM track. … If the survey paper will naturally be rejected by COLM2024, we will withdraw the submission accordingly.
>
> > A: Please address this in your rebuttal. We don't have a specific policy about survey papers, and **they will be judged according to the contribution**, similar to conferences like ICLR/NeurIPS and ACL venues.   -Yoav
>
> As noted by the PC, survey papers are also valued as a form of contribution by COLM'24, similar to other top-tier conferences like ACL, which even categorizes surveys as a distinct type of contribution on its submission site. One can find survey papers such as "Reasoning with Language Model Prompting: A Survey" accepted by prestigious venues like ACL/EMNLP/NAACL. Unlike well-defined reasoning tasks, Strategic Reasoning with LLMs remains unorganized and lacks systematic consolidation despite numerous related efforts. Our paper precisely provides a timely and comprehensive review, contributing to the community by standardizing research efforts in this field.
>
> **Response to "What major challenges and opportunities exist in applying LMs to strategic reasoning scenarios, and which areas are receiving more or less attention in LLM research?"**
>
> In our paper, we highlighted three key benefits of utilizing LLMs in the introduction. Regarding the replacement of traditional methods, we acknowledge that LLMs have not yet reached a stage where they can outperform state-of-the-art RL models in domains such as poker or Go. However, LLMs have shown advantages in novel research areas like conversational games (e.g., Diplomacy), where traditional RL methods may fall short. LLMs have broadened the application scenarios of strategic reasoning, presenting new challenges and opportunities.
>
> We aim to offer significant value through comprehensive synthesis and critical analysis of existing works, alongside our in-depth insights on this direction. To address your concern about the depth of discussion, we will expand the "scenarios" section and delve into specific challenges and opportunities across scenarios, explaining the motivation for using LLMs and assessing the prospects for success.
>
> Thank you for your insightful suggestions. We look forward to further discussions with you.

---

### Official Review · Reviewer_ppn3 · 2024-05-14

**Rating:** 6
**Confidence:** 3
**Ethics Flag:** 1

**Summary:**

This paper provides a survey regarding strategic reasoning with LLMs. The paper well motivated why LLMs are game changers for strategic reasoning and frame the existing works according to application, methods, and evaluation. Interesting challenges and insights are discussed at the end.

**Questions To Authors:**

I agree with the point that we should have a unified framework and benchmark. I am curious how could we define any proposed benchmark as unified and standard enough so that every related research should evaluate with?

**Reasons To Accept:**

- Strategic reasoning is an important topic. Though many existing works (especially relying on LLMs) are published recently, their effort indeed widely spread in terms of topics and evaluation criteria. Thus, it becomes hard to follow and compare across papers. The survey paper calls out a unified framework and benchmark, which is crucial and necessary for the related research community. Researcher should align on the same standard.

- The survey provides good angles to well unify most of the related works.

**Reasons To Reject:**

- As a survey paper, the paper has limited novelty.

- Some points on outlook section can go deeper with more interesting insights and designs among existing work.

---

> ### Author Rebuttal · Authors · 2024-05-28
>
> Thank you for your careful review and valuable comments on our work.
>
> **Response to Question "How could we define any proposed benchmark as unified and standard enough so that every related research should evaluate with?"**
>
> We believe a unified benchmark should embody three key characteristics:
>
> 1. **Well-Controlled Environments:** The benchmark should operate within controlled environments with minimal randomness, facilitating reproducibility and ease of experimentation.
> 2. **Reasonable (Objective or Subjective) Evaluation:** As discussed in section 5, a unified benchmark for strategic reasoning with Large Language Models (LLMs) should be capable of objectively or subjectively assessing their strategic reasoning abilities. This includes evaluating long-term planning capabilities and dynamic adaptability, among other facets of strategic reasoning.
> 3. **Scalability Across Scenarios:** The benchmark should be adaptable to a broad spectrum of strategic reasoning scenarios, enabling researchers to explore diverse applications and challenges.
>
> As mentioned in our paper, some benchmarks derived from classical game-theoretical problems for strategic reasoning lack higher-level abstraction, which limits their scalability and applicability to real-world applications. One feasible approach involves building upon existing RL-based environments, such as Robot Warehouse (RWARE)[1] and the StarCraft Multi-Agent Challenge [2]. These environments offer well-defined and controllable settings, laying a solid foundation for benchmarking strategic reasoning abilities. However, further refinement is necessary to ensure the benchmark meets the criteria of reasonable and comprehensive evaluation and scalability.
>
> By leveraging established environments as starting points, we can enhance the benchmark's capacity to assess LLMs' strategic reasoning capabilities while fostering innovation and advancement in the field. Thank you for prompting this critical discussion, and we welcome any additional insights or suggestions you may have.
>
> [1] F. Christianos, L. Schäfer, and S. V. Albrecht. Shared experience actor-critic for multi-agent reinforcement learning. 2020.
>
> [2] M. Samvelyan, T. Rashid, C. S. de Witt, G. Farquhar, N. Nardelli, T. G. J. Rudner, C.-M. Hung, P. H. S. Torr, J. Foerster, and S. Whiteson. The StarCraft Multi-Agent Challenge. CoRR, abs/1902.04043, 2019.

---

### Decision · Program_Chairs · 2024-07-10

**Decision:**

Accept

**Comment:**

This paper provides a survey on strategic reasoning with LLMs. Most reviewers appreciated the comprehensive and timely nature of the survey, recognizing its importance in the burgeoning field of LLM-based strategic reasoning. This paper received mixed reviews but after clarifying the suitability of survey papers in COLM, one reviewer raised their score and all reviewers reach a consensus on this work. The paper was commended for its clear organization, thorough literature review, and insightful future research directions. Concerns regarding deeper analysis of related work and more specific discussion of unique challenges and evaluation metrics within different scenarios seem to be mostly addressed in the rebuttal. Overall, the consensus leaned towards acceptance, emphasizing the paper's value in systematizing research efforts and contributing to the strategic reasoning community.